# Designing Composite Stimuli-Responsive Hydrogels for Wound Healing Applications: The State-of-the-Art and Recent Discoveries

**DOI:** 10.3390/ma17020278

**Published:** 2024-01-05

**Authors:** Anna Michalicha, Anna Belcarz, Dimitrios A. Giannakoudakis, Magdalena Staniszewska, Mariusz Barczak

**Affiliations:** 1Chair and Department of Biochemistry and Biotechnology, Medical University of Lublin, Chodzki 1, 20-093 Lublin, Poland; 2Department of Chemistry, Aristotle University of Thessaloniki, 54124 Thessaloniki, Greece; 3Institute of Health Sciences, Faculty of Medicine, The John Paul II Catholic University of Lublin, Konstantynów 1J, 20-708 Lublin, Poland; 4Institute of Chemical Sciences, Faculty of Chemistry, Maria Curie-Sklodowska University, 20031 Lublin, Poland

**Keywords:** hydrogels, wound dressings, wound healing, controlled release, stimuli-responsive, drug delivery strategies

## Abstract

Effective wound treatment has become one of the most important challenges for healthcare as it continues to be one of the leading causes of death worldwide. Therefore, wound care technologies significantly evolved in order to provide a holistic approach based on various designs of functional wound dressings. Among them, hydrogels have been widely used for wound treatment due to their biocompatibility and similarity to the extracellular matrix. The hydrogel formula offers the control of an optimal wound moisture level due to its ability to absorb excess fluid from the wound or release moisture as needed. Additionally, hydrogels can be successfully integrated with a plethora of biologically active components (e.g., nanoparticles, pharmaceuticals, natural extracts, peptides), thus enhancing the performance of resulting composite hydrogels in wound healing applications. In this review, the-state-of-the-art discoveries related to stimuli-responsive hydrogel-based dressings have been summarized, taking into account their antimicrobial, anti-inflammatory, antioxidant, and hemostatic properties, as well as other effects (e.g., re-epithelialization, vascularization, and restoration of the tissue) resulting from their use.

## 1. Introduction

The major failure that medicine faces in terms of wound dressings is that despite their undeniable advances, they are still not as efficient as they are expected to be. Complications associated with the healing of wounds that are not cured with the currently used wound dressings still cause many global health risks as well as economic concerns. Wound-related complications affect over six million people annually in the United States, incurring a cost of USD 25 billion [1]. Chronic wounds are predominantly observed in the elderly, affecting approximately 3% of the U.S. population aged 65 and above. The aging demographic trend suggests that by 2060, over 77 million elderly individuals in the United States may deal with persistent open wounds. The advanced wound care market worldwide is expected to achieve a value of USD 18.7 billion by the year 2027, demonstrating a Compound Annual Growth Rate (CAGR) of 6.6% from 2020 to 2027 [2]. Global estimates indicate that approximately 463 million adults are currently living with diabetes, and this figure is anticipated to surge to USD 700 million by the year 2045 [2,3,4]. This is particularly relevant, as diabetes patients are at an increased risk of developing diabetes-related wounds. The global market for diabetic foot ulcers anticipates a favorable CAGR of 6.8% from 2019 to 2026, with a potential valuation of USD 11 billion by the end of 2026 [2]. Moreover, the global market for venous ulcers (VU) treatment is anticipated to reach USD 4.8 billion by 2026, with an annual growth rate of 6.4% from 2019 to 2026 [2,5]. Based on the data presented above, it is evident that there is an urgent need for the development of novel and effective wound dressing materials that meet the specific requirements in the treatment of different types of wounds.

Currently used commercial wound dressings show many limitations such as a lack of specific responsive properties to the changes in the environment. They are also usually applied in a one-for-all wounds manner [6]. Due to the high demand for effective wound care methods in the field of medicine, there is a growing pressure to create advanced materials tailored to accelerate the healing process in various clinical contexts [7,8]. Ideally, dressing materials are meant to protect the wound against external factors, support the epithelial renewal process, prevent bacterial infections and further damage of skin integrity, as well as to ensure adequate moisture of the wound environment [9]. Properties of available dressing materials, both the newest and the conventionally used ones, are limited by their rapid degradation, poor adhesion, inefficient exudate absorption, lack of therapeutic release properties, and inability to prevent protein adhesion to the wound dressing surface [10,11,12].

In the case of uncomplicated wounds, the healing process occurs in the form of a sequence of interrelated phases. These phases include hemostasis, inflammation, proliferation, and remodeling, often occurring concurrently and overlapping [13]. Depending on the consequences and pathogenesis, the wound can be qualified as acute or chronic. The primary distinction between them lies in the biochemical microenvironment and concentration of specific components within the wound site, which is schematically presented in Figure 1. Acute wounds result from trauma or surgical procedures and undergo a healing process over a specific period of time, usually within a few weeks [14,15]. They are often associated with common injuries such as cuts, lacerations, abrasions, burns, and surgical or accidental incisions [16,17]. Chronic wounds such as diabetic foot ulcers, venous leg ulcers, ischemic ulcers, pressure ulcers, and many more, are frequently associated with long-term health disorders. One of the most prevalent reasons for chronic wound occurrence is the improper treatment or infection of acute wounds [18,19]. The chronic wound healing process is delayed due to various factors. Those factors include aging, the stage of diabetic disease, medication compliance, associated peripheral neuropathy, immunocompromised status, as well as arterial and venous insufficiency [1]. In the case of chronic wounds, the inflammatory phase is prolonged [20,21]. Increased levels of inflammatory mediators can impair the functioning of growth factors and the extracellular matrix, both of which are crucial for the healing process [22,23]. They complicate the healing process and extend it up to three months. Additionally, chronic wounds often undergo bacterial infection, which can extend the inflammatory response [20,21]. It should also be noted that acute wounds, usually undergoing the above-mentioned healing stages within defined timeframes, can also be infected, which negatively affects the prognosis of their treatment [16,17].

The intact skin acts as a physical barrier, preventing bacteria and other pathogens from invading internal tissues. However, when the skin is cut, scraped, or otherwise damaged, it loses its protective integrity. The damage to the skin’s protective barrier offers an opportunity for microbial invasion [24,25]. Wound infections, similar to the earlier mentioned inflammations, decrease the activity of growth factors. In addition, fibrin/fibrinogen is a target of bacterial proteins due to its role in defense against bacterial infection [20,21,26]. Bacterial infections not only have the potential to significantly slow down the wound healing process, but also pose a substantial risk of inflicting severe damage to tissues and cells. In extreme cases, they can even be life threatening.

Interactions between the wound environment and bacteria include the contamination, colonization, local infection, and finally spreading of the infection and the emergence of a chronic wound condition [1]. As mentioned above, chronic wounds are frequently associated with bacterial infections, which can slow down the process of angiogenesis, due to the release of tissue-destroying (lytic) enzymes, exotoxins, and endotoxins; all of which potentially lead to the deterioration of wound healing [27,28]. Overall, bacterial infection can result in an imbalance of regulatory molecules that are crucial for the healing process, leading to impairment and a retention in tissue repair at one of the mentioned-before healing stages [29].

Historically, the primary approach to preventing and treating wound infections has involved the use of antibiotics. Conventional antibiotics (tetracyclines, aminoglycosides, quinolones, and cephalosporins) have traditionally been noted as successful in killing bacteria by disrupting their cell walls and interfering with essential processes such as protein and nucleic acid synthesis [30]. However, over the last few decades it has been shown that overuse or inappropriate utilization of these substances lead to consequences such as the emergence of multidrug-resistant strains with a propensity for bacterial biofilm formation [31,32]. The last is an important phenomenon because drugs usually cannot penetrate the bacterial biofilm structures [30]. Because of this, traditionally used pharmaceuticals and antibiotics have demonstrated reduced efficacy when compared to initial expectations [7,29,33]. Therefore, researchers and healthcare practitioners actively work on pioneering therapeutic strategies to prevent and manage infections in both acute and chronic wounds [1,2].

Numerous types of wound dressings have been developed to facilitate wound healing, including gauzes, transparent films, foam dressings, hydrogels, hydrocolloids, and hydroconductive dressings [34]. They are schematically presented in Figure 2. An ideal wound dressing should: (i) be characterized by a high biocompatibility and lack of toxicity, (ii) have adequate durability/mechanical properties, (iii) promote cell adhesion and differentiation, (iv) provide constant moisturization, (v) adhere well to the wound tissue but at the same time should be easy and painless to remove, (vi) ensure optimal gas exchange between the wound and the surroundings, and (vii) exhibit remarkable antimicrobial action [35].

Gauzes are the oldest and most economical, readily available, and highly absorbent traditional wound dressings. They can easily conform to the shape of the wound and they are widely used for dressing both infected and non-infected wounds with a significant amount of exudate. However, gauze dressings are not ideal for wounds, as upon removal, they may cause trauma, mechanical injury of the healing wound, and consequently, patient discomfort [34,36]. Transparent film dressings represent a refined progression from traditional gauzes, offering the capacity to maintain a moist wound environment, facilitate gas exchange, and protect against external bacterial contamination. Their easy adaptability and pain-free removal set them apart. Nonetheless, their lack of swelling capability makes them unsuitable for highly exudative or infected wounds, where the coexistence of exudate and infection can potentially exacerbate bacterial proliferation [34,36,37]. Foam dressings are recommended for managing wounds with high levels of exudate. This is associated with their outstanding ability to absorb substantial fluid volumes while providing thermal insulation and facilitating gas exchange. Due to its impressive absorbent capability, this type of dressing can be changed every seven days in non-infected wounds, while daily changes are recommended in the presence of infections [38,39]. Hydrocolloids are gel-forming systems made of an elastic matrix with hydrophilic polymers and they absorb fluids. They efficiently seal the wound bed without needing additional dressings. Hydrocolloids speed up healing by enhancing autolysis and debridement. However, they are not suitable for infected wounds due to their occlusive nature and can cause trauma during removal [34,36,40]. A hydroconductive dressing features a specific multilayer structure that allows for the absorption of wound exudate, removal of debris from the wound bed, and subsequent transport of these by-products into its core [34,36,41].

Among the wound dressings, hydrogels have emerged as the most promising candidates for wound dressings [42]. This is primarily due to their excellent hydrophilicity, biocompatibility, and three-dimensional porous structure that resembles the extracellular matrix (ECM) [43]. Compared to traditional dressings, hydrogels often exhibit better therapeutic effects on wounds that are prone to bacterial infections [44]. As a result, hydrogels have garnered significant attention among researchers for their tremendous promise in wound healing and formulation of wound dressings. This is related to a number of very desirable and often unique properties exhibited by hydrogel-based systems [45]. Firstly, the three-dimensional architecture of the hydrogel constructs favors the formation of an environment that fosters proper regeneration, acting as a framework for regeneration and healing processes. Additionally, their mechanical strength matches the native tissue to provide a highly bio-mimicked environment for better cell attachment, spreading, and proliferation. Moreover, the morphology of hydrogels mimics the morphology of the extracellular matrix and macromolecules, while hydrogel porosity provides effective cell infiltration and enhances transport of the various species needed for wound healing. It is also noteworthy that they are characterized by the capability for effective water retention that provides a humid environment to display normal cell behavior (e.g., proliferation). Additionally, hydrogels can excessively absorb wound exudates, limiting the microbial growth near the wound. In this area stimuli-responsive composite hydrogels containing antibacterial agents attracted significant attention.

There is already a wide range of commercially available hydrogel-based dressings on the market, which are summarized in several papers [46,47,48]. However, there is still a lot of work and research going on to produce better and more effective hydrogel dressings (particularly stimuli-responsive hydrogels) and address the challenges associated with them, which are discussed in the next section.

## 2. Challenges Related to Hydrogel-Based Wound Dressings

Due to the combination of high water content, softness, flexibility, biocompatibility, and bioactivity, hydrogels have gained massive popularity in the wound dressings field. Their structure and properties make them dedicated formulations to overcome many complications related to the natural characteristics of chronic wounds [49]. These include the appearance of excessive exudates, massive bleeding, and persistent bacterial infections.

The hydrogel formulation, due to its high liquid absorption capacity, can usually absorb the excessive wound exudates. In the normal wound healing process, the exudate levels typically decrease over time. However, in the case of chronic wounds, such as various types of ulcers, exudate production is often excessive due to the ongoing inflammatory process [50,51]. Hydrogels’ capacity to efficiently absorb exudate plays a vital role in managing chronic, bacterial infected wounds [52,53]. As a desired side effect, hydrated wound dressings help to control the temperature of the wound-affected area, accelerating the healing process [54,55,56,57,58,59].

Unregulated and massive bleeding resulting from trauma can give rise to a range of issues, including hypothermia, lowered blood pressure, susceptibility to bacterial infection, and even the onset of shock [60]. Hydrogels are a suitable platform for producing bleeding-controlling materials due to their exceptional liquid absorbing properties, which enable them to eliminate the excess of blood from the wound. Moreover, some hydrogels may be supplemented with substances promoting blood clot formation, including tannic acid [61,62,63], zeolite and kaolin [64,65,66], or polydopamine [67,68]. Such materials can actively and rapidly stop bleeding in wounds, promote clot formation, and minimize blood loss. This facilitates wound healing processes and reduces the risk of complications associated with uncontrolled bleeding [62,69,70,71].

Moreover, hydrogel wound dressings represent a promising platform for modifications using antimicrobials [68,72,73]. Antimicrobial-loaded hydrogels act as biological shields that hinder harmful elements such as bacteria from infiltrating the wound. They also show the ability to release therapeutic agents that aid the wound healing process in response to varying environmental conditions [45]. This is important in light of the information we provided in the previous chapter of this work.

Regardless of the factors resulting from the wounds themselves, hydrogels must also meet other conditions, such as good mechanical properties, which include both resistance to compression and resistance to stretching [74,75]. These properties allow the hydrogels to withstand heavy loads and strains during their application in critical places of the body, such as the joints or neck. First-generation hydrogels, when damaged (e.g., due mechanical compression), could easily lose their initial mechanical properties and their network structure could be affected, resulting in a reduction in their lifetime [76,77]. This appears because destroyed hydrogels cannot self-heal and reform the broken bonds. Self-healing hydrogels are a potential solution to this problem because they have a built-in ability to autonomously repair their original properties and structure in response to damage [78]. It should be always remembered that natural hydrogels are more susceptible to enzymatic degradation than their synthetic counterparts (e.g., polyethylene glycol, polyvinyl alcohol, poly(N-isopropyl acrylamide)), which exhibit slower degradation rates. Therefore, natural hydrogels are usually coupled with various crosslinkers, bioactive compounds, antibiotics, or metal ions to achieve the enhanced performance [79].

The above-mentioned problems occurring during wound treatment (excessive exudates, blood flow, infections, the need for the dressing to adapt to the wound) can be solved by new generation hydrogels that react to stimuli occurring in the environment. Such hydrogels can respond to problems that arise during wound healing in response to internal (pH, wound temperature, the presence of free radicals) or external (near infrared (NIR)) stimuli [80,81]. This response may be the release of active factors (e.g., antibacterial substances), a change in absorption capacity or a change in structure, transparency, gel volume, mechanical characteristics, or surface properties [82,83]. Such incredibly versatile structures are referred to as “stimuli-responsive”, “intelligent”, or “smart” hydrogels [84,85]. This responsiveness allows for the controlled release of active substances, making them highly effective in wound management [33,86,87]. The development of such intelligent carriers that can adapt to internal conditions and prevent a bacterial infection is the big challenge in wound regenerative medicine. The advantages of stimuli-responsive hydrogels are graphically summarized in Figure 3.

When discussing materials responsive to stimuli, it is also worth mentioning shape memory polymers (SMPs). SMPs are a promising tool for designing new generation biomaterials for medical applications due to their unique capability of memorizing their initial shape and reverting to it from a deformed state when exposed to one or more suitable external stimuli, such as heat, light, pH, electricity, a magnetic field, and moisture [88]. In recent years, there has been a growing interest in these polymers in scientific research connected to biomedical applications [89,90,91]. For example, Panda et al. incorporated p-coumaric-acid-modified water-soluble chitosan (M-Cs) into the poly(vinyl alcohol) (PVA) polymer matrix. Compared to the PVA control variant, the addition of M-Cs resulted in improved water-induced shape memory behavior of the material, achieving a shape recovery ratio close to 100% [88].

## 3. Various Stimuli Triggering the Hydrogels’ Response

Numerous stimuli-responsive hydrogels possess antibacterial, anti-inflammatory, or antioxidative properties [92,93]. Some even demonstrate dual properties, combining two of the abovementioned characteristics simultaneously [61,94,95,96,97], thus making them favorite candidates for the development of self-regulated drug delivery systems [98]. Currently, stimuli-responsive composite hydrogels containing antibacterial agents attracted the highest attention in scientific society and constitute a significant number of ongoing studies [99]. Therefore, in this review we focused on the stimuli-responsive hydrogels containing a wide range of antibacterial additives as an alternative to often unreliable antibiotics, in particular polyphenols, antibacterial polypeptides, and silver nanoparticles. Stimuli-responsive hydrogels can be categorized as [100]:non-contact stimuli-responsive hydrogels (e.g., light-responsive, thermo-responsive, magnetic/electric field-responsive),contact stimuli-responsive hydrogels (e.g., pH-responsive, ion-responsive, chemically/biochemically responsive),multistimuli-responsive hydrogels (susceptible to the simultaneous or sequential action of two or more stimuli).

Stimuli identified as the most promising and efficient for controlling behaviors of the resulting hydrogels are: pH, reactive oxygen species (ROS), temperature, and NIR, as schematically presented in Figure 4. This review focuses on these four types of stimuli as the most frequently used both in research and in medical translational studies. This, of course, does not limit the wide range of possibilities for using other types of stimuli, such as ionic strength for example, which can be precisely used not only to induce hydrogelation but also to modulate hydrogel properties (e.g., mechanical properties) and response (e.g., transport properties) [101,102,103].

### 3.1. pH-Responsive Hydrogels

pH is a tightly regulated factor that plays a crucial role in maintaining the proper function of the skin and its levels vary across different skin layers, increasing from the surface to the deeper layers, therefore pH tends to fluctuate during differing stages of the wound healing process [104]. pH is as an indicator of the wound’s condition, and changes in pH levels can be used to predict whether a wound is likely to heal or deteriorate [105]. The pH levels differ based on the condition of the skin: healthy skin typically maintains a slightly acidic pH (5–6), acute wounds exhibit a pH around 7.4, while chronic wounds tend to have a more alkaline pH, ranging from 7.3 to 10, partly due to the presence of proliferating bacterial colonies [106]. These pH fluctuations can impact bacterial infection as well as colonization, which are common characteristics of chronic wounds.

The appropriate regulation of wound pH during the different healing phases can accelerate wound healing. Restoration of the acidic environment in chronic wounds can reduce microbial colonization on the skin surface and enhance adipose tissue metabolism. However, the proliferation and migration of keratinocytes and fibroblasts favor a slightly more alkaline environment with a pH of around 8.3. For example, creating an acidic environment in the initial stages (hemostasis and inflammation) can inhibit bacterial infection and promote vascular regeneration, whereas the alkaline environment (observed during wound hyperplasia and remodeling) can enhance cell proliferation and skin remodeling [104]. Nevertheless, continuous wound pH regulation throughout all healing phases remains a challenging goal for scientists [107].

pH-responsive matrix degradation is one of methods designed for the control of drug release from hydrogels. The desired situation is, as bacteria multiply, this degradation is activated. During this phase, the strength of the hydrogel’s three-dimensional structure diminishes, leading to the formation of larger internal pores that in consequence enables the release of the drug to inhibit the bacterial growth. Once the pH of the wound returns to its typical level, the above process stops [108].

This strategy was implemented by Bonetti et al. in the production of a methylcellulose-based hydrogel, which was crosslinked using ester bonds. Ester bonds are susceptible to hydrolysis in an alkaline environment, causing the hydrogel network to expand and facilitate drug release [15]. The change in swelling behavior can lead to the mentioned above shift in the pore size within the hydrogel structure used for pH-responsive drug delivery or signal transmission [109,110]. This method has also been reported in other studies related to pH-responsive hydrogel wound dressings. The pH-dependent alterations in hydrogel size predominantly concern the way these materials expand or contract [111,112].

To sum up, pH-responsive wound dressing hydrogels offer unique benefits in wound care due to their exceptional biochemical and mechanical characteristics [113]. They effectively assess the wound’s condition by monitoring its pH, facilitate and accelerate the controlled wound healing, and lower the risk of infection. Such constructs have the capability to adjust their structure and active substances release in response to a pH stimulus [114,115].

### 3.2. ROS-Responsive Hydrogels

Reduction–oxidation (redox) potential is a biological parameter that can be influenced by various factors and may undergo alterations in certain states of disease, such as inflammation, cancer, or hypoxia [116]. ROSs are pivotal regulatory elements in the wound healing process, facilitating natural skin repair. They also play a significant role in oxidative bacterial elimination, thereby promoting angiogenesis and re-epithelialization at the wound site [117]. However, excessive ROS levels can lead to oxidative damage, hindering the proper healing process. It causes an inflammatory response that inhibits the functions of both endogenous stem cells and macrophages and also retards tissue regeneration [118,119]. As a consequence, the wound remains in the inflammatory phase for a long time. Therefore, the extended healing time does not allow for a smooth transition into the proliferation and remodeling phase [120]. Notably, hypoxia impedes the wound healing process by inhibition of fibroblast proliferation and collagen production. The overproduction and accumulation of ROSs in the wound environment is also the common reason for the emergence of bacterial infection or even a diabetic wound state. This is particularly evident in chronic wounds, where prolonged oxidative stress prevails. Clearly, the improvement in the healing process can be achieved by reducing the bacterial presence [20].

Hence, antioxidant hydrogels have the potential to reduce the excessive ROSs in wound sites by neutralizing free radicals, interrupting the free radical chain reactions, and alleviating dysfunction in the immune system [121]. Targeting oxidative stress in chronic wounds through the restoration of the redox equilibrium has demonstrated effectiveness in enhancing the proper wound repair [122]. The development of antioxidant hydrogels can be approached in two main ways: either by using hydrogels as carriers for ROS scavengers or by creating hydrogels with inherent antioxidant properties, which can be achieved by using antioxidant macromolecules as hydrogel precursors. Considering this evidence, investigations into designing antioxidant wound dressing materials aimed to assess their impact on the healing process acceleration. This can be achieved by hydrogel loading with natural polyphenols such as tannins, gallic acid, and curcumin to capture and neutralize free radicals [22,46,123].

In the case of hydrogels responsive to ROSs within the wound environment, the current strategy applies either the introduction of an ROS-responsive block into the backbone of a hydrogel-forming polymer or the use of the polymers with ROS-responsive side chains [124]. An increase in ROS concentration in the surrounding environment leads to the hydrolysis or degradation of chemical bonds in the hydrogel with a hydrophobic-to-hydrophilic transition or polymer chain scission. This in turn leads to the controlled release of hydrogel-loaded drugs. As the level of ROSs varies depending on successive phases of the healing process [125], ROS-responsive hydrogels also release therapeutic substances in response to ROS fluctuations occurring during these phases [126]. Thus, this controlled release of therapeutic substances underlines their responsiveness to the dynamic environment of the wound.

Taking these facts into consideration, ROS-responsive biomaterials (RRBs) are gaining growing potential for mitigating oxidative stress in tissue microenvironments and serving as targeted carriers for drug release in response to physiological oxidative conditions in the wound environment. In particular, RRBs show a real potential in difficult wound management, including diabetic wound treatment, which is one of the greatest challenges of 21st century medicine.

### 3.3. Temperature-Responsive Hydrogels

Temperature plays a crucial role in wound healing as it affects the rates of enzyme responses, given their temperature-dependent nature. Additionally, temperature serves as a conventional indicator in the clinical assessment of chronic wounds, reflecting classic signs and symptoms [127]. The temperature could be a signal for specialized wound dressings to activate or respond in a particular way. Temperature-responsive hydrogels have significant potential in drug or cell delivery systems and injured tissue repair. They are commonly used in designing responsive systems due to their ease of control [128].

Hydrogels sensitive to temperature changes undergo the transition between a liquid and solid (adhesive) state, displaying enhanced flexibility for conforming to irregular wound surfaces [129]. Temperature-responsive hydrogels are usually formed using non-covalent interactions between the components, and their physical state depends on temperature. This phenomenon can trigger the sol–gel transition, enabling the gel to adapt perfectly to the wound site, in particular for injectable gel types. Incorporation of the drugs into the liquid-state hydrogel ensures their homogeneous dispersion. Meanwhile, the rapid gel formation (via the sol–gel transition, often occurring at a physiological temperature) prevents an initial drug burst release, providing the sustained delivery of active substances. Such materials can swell and shrink according to the environmental temperature [130], with related changes in volume due to the hydrophobic/hydrophilic functional groups present in the hydrogel structure [131,132]. This approach simplifies the use of hydrogels, making the therapeutic process straightforward and user-friendly [133,134].

Normal body temperature is approx. 37 °C. In relation to this fact, hydrogel wound dressings are often composed of thermal-sensitive materials with a critical solution temperature lower than body physiological temperature. So-called lower critical solution temperature (LCST) hydrogel can shrink when the temperature is above the LCST point. Hydrophobic moieties, such as propyl, ethyl, and methyl groups, are characteristic for temperature-responsive hydrogels. The polymers with LCST, which are currently used in biomedical applications, are as follows: polyethylene glycol (PEG) (106–115 °C), poly(propylene glycol) (PPG) (10–40 °C), poly(vinyl alcohol) (PVA) (125 °C), poly(N-isopropylacrylamide) (PNIPAAM) (32 °C), poly(methyl vinyl ether) (PMVE) (28–34 °C), poly(N-vinyl caprolactam) (PNVCa) (30–50 °C), and Pluronic-F127 (PF-127) (26.5 °C) [135,136]. Chitosan (CH), chondroitin sulfate, hyaluronic acid (HA), alginate (Alg.), dextran, and cellulose belong to natural polymers that can be blended with thermosensitive hydrogel polymers. This opens the possibility for the development of innovative hydrogels with favorable characteristics suitable for tissue engineering applications [83,134,137].

Thermosensitive hydrogels with positive characteristics have the ability to increase their solubility above the upper critical solution temperature (UCST). Polymers exhibiting UCST behavior become less soluble at temperatures exceeding their critical point, leading to a sol–gel transition. In contrast, hydrogels with a lower critical solution temperature (LCST) contract and precipitate from the solution above their critical temperature. This phase transition is often reversible, allowing for controlled changes in the hydrogel state. Thermosensitive hydrogels with LCST behavior swell in response to a decrease in temperature [80,138].

### 3.4. NIR-Responsive Hydrogels

The near-infrared (NIR) light, falling within the wavelength range of 780–1700 nm, is considered a therapeutic window for light-activated delivery systems in vivo [139]. The distinctive feature of NIR light is its wide range of light wavelengths (780–1700 nm). This feature makes the NIR-responsive materials excellent candidates for use as therapeutic agents and biological tools in a variety of biomedical applications [140,141]. NIR’s ability to penetrate tissues deeply with minimal phototoxicity and non-invasiveness is the big advantage in biomedicine [142,143,144,145].

Light-responsive materials, particularly those responding to NIR light, have emerged as a highly promising approach for managing bacterial infections during the wound healing process [146,147,148,149]. Scientists have shown the significant role of NIR stimulation in the development of wound dressing hydrogels with enhanced antibacterial properties [150,151,152,153]. The use of near infrared (NIR) in wound treatment offers benefits such as accelerating the healing process by stimulating collagen production [154], improving blood microcirculation, anti-inflammatory and bactericidal effects, and effectiveness in the treatment of chronic wounds [155].

Under NIR light the structure of materials changes as a result of chemical bonds breaking and changes in molecular structure, thus enabling the release of active substances. In order for a biomaterial to react to NIR by generating heat, it must contain factors responsive to NIR. Such materials containing NIR-responsive additives may lead to thermal decomposition of the hydrogels and, consequently, the release of active substances enclosed in their structure [148,156,157]. When the drug enclosed in a hydrogel structure is in the form of NIR-sensitive nanoparticles, heating the constructs may result in a faster controlled release. It is possible due to a change in the hydrogel structure, resulting in increased drug permeability. Such photothermal agents include gold nanoparticles [158], silver nanoparticles [159], or carbon nanomaterials [160], which can produce heat when exposed to NIR light, facilitating drug release [161].

It is worth mentioning that NIR stimulation can be used to obtain other effects besides targeted drug release. For example, it is also an amazing tool for the synthesis of self-removable wound dressings. For example, Zhao et al. [162] designed hydrogels consisting of catechol–Fe^3+^ coordination cross-linked poly(glycerol sebacate)-co-poly(ethylene glycol)-g-catechol and quadruple hydrogen bonding cross-linked ureido-pyrimidinone-modified gelatin. Upon exposure to near-infrared (NIR) light, the hydrogel exhibited photothermal effects. The rise in temperature caused the breakdown of hydrogen bonds, leading to the dissolution of the hydrogel and facilitating its easy removal from the skin wound.

### 3.5. Examples of Existing Stimuli-Responsive Hydrogels

Depending on the components used in their production process, the resulting constructs exhibit distinct characteristics and properties. Examples of various stimuli-responsive wound healing constructs were presented in Table 1 to highlight the huge diversity of such technologies in wound treatment applications. The more systematic discussion is provided in Section 4.

## 4. Loading Stimuli-Responsive Hydrogels with Active Substances

Stimuli-responsive hydrogels can be classified not only based on the specific stimulus they respond to but also on the type of active substance they are loaded with. Both the hydrogel matrix and the loaded agent determine the final physiochemical and healing properties of the resulting biomaterial, which have been concisely summarized in Table 2. The dominant group of active substances in this type of hydrogel is compounds showing antibacterial activity, with particular emphasis on substances other than antibiotics. This trend is based on the search for alternatives to drugs that a significant number of bacterial strains have developed resistance to. Nowadays, polyphenols, antibacterial polypeptides, and silver nanoparticles are used most frequently for the preparation of stimuli-responsive hydrogels containing non-antibiotic substances. Below, the representatives of such hydrogels are described, taking into consideration various stimuli used for the liberation of their activity. At this point it is worth mentioning that instrumental techniques and modeling approaches employed to monitor stimuli-induced variations play an important role; the interested reader is therefore referred to a number of interesting publications in this field [100,170,171,172].

### 4.1. Hydrogels Loaded with Polyphenols

Polyphenols are a diverse group of natural compounds that are widely distributed in plants, marine organisms, and various other sources. They have gained increasing attention in biomedical fields for their numerous potential health benefits, including their inherent biocompatibility, antioxidant and antibacterial activities [199]. Polyphenol compounds include functional groups such as catechol and pyrogallol, enabling them to engage with a multitude of molecules through the formation of diverse non-covalent interactions (e.g., hydrogen bonding, π–π interactions, cation–π interactions, etc.) as well as covalent interactions (e.g., Michael addition/Schiff-base reaction, polyphenol–metal coordination, etc.) [200,201]. It makes them great candidates for the synthesis and modification of hydrogels’ biomaterials for medical purposes, including wound dressings (WDs).

The addition of polyphenols to polymers used for hydrogel WD synthesis allows for numerous interactions between these two compounds, leading to a tighter and more interconnected network. This contributes to improved mechanical strength and structural integrity of the matrix. Consequently, they often exhibit exceptional properties such as adhesion, high elasticity, and self-healing, qualities that are highly desired in the design of “intelligent” hydrogels. These properties are particularly desired during the healing of wounds with continuous and persistent bleeding because they ensure excellent hemostatic performance even with deep wounds after adhering to the wound site [202,203]. Introducing polyphenols into the hydrogel structure enhances the overall performance of hydrogels.

Tannic acid (TA), belonging to the class of polyphenols, is one of the most widely recognized within this category. TA contains many hydroxyl groups showing the affinity for the formation of hydrogen bonds with proteins and other biomolecules. This property allowed for the use of TA for traditional medicine to treat a variety of maladies [204]. TA has been shown to reduce inflammation as an antioxidant and can induce apoptosis in several cancer types. TA has also displayed antiviral and antifungal activity. Moreover, taking into account the results of new preclinical and clinical studies and the growing resistance of bacteria to antibiotics, new intriguing perspectives emerge for this natural compound in relation to TA application as an antibacterial agent. In biomaterials research, as was already mentioned, TA can enhance the mechanical properties of natural and synthetic hydrogels and polymers due to its crosslinking property, imparting beneficial attributes to these materials [204,205].

Although other polyphenols, such as gallic acid and resveratrol have been employed in the production of stimuli-responsive hydrogel dressings, TA stands out as the most effective in this regard. The comparative analysis presented in Table 3 illustrates examples of hydrogels synthesized using various polyphenols; however, it is evident that TA is superior to others in terms of its suitability for the production of stimuli-responsive hydrogel dressings.

The gelatin-TA (GelTA) hydrogel accelerates skin healing by releasing TA in a pH-dependent manner. This release is responsible for an antibacterial, antioxidant, hemostatic, and anti-inflammatory activity of the hydrogel. Moreover, the GelTA hydrogel significantly enhances extracellular matrix formation, wound closure, re-epithelialization, and collagen deposition in vivo by offering cell adhesion sites within the gelatin matrix [70]. Ni et al. synthesized TA-conjugated nanoparticle hydrogels (PPBA-TA-PVA) by mixing TA, phenylboric-acid-modified polyphosphazene (PPBA), and poly(vinyl alcohol) (PVA). PPBA–TA–PVA hydrogels were shown to be a promising platform for reducing inflammation and speeding up wound healing. They exhibited ROS-scavenging activity due to the ROS-responsive degradation depending on phenylboric acid presence. As a result of this degradation, the hydrogel released TA, which was responsible for the ROS-scavenging phenomenon. This led to shortening the healing time of diabetic wounds [173]. Pan et al. [123] synthesized injectable hydrogel with self-healing properties and antibacterial activity against *Staphylococcus aureus* and *Escherichia coli* based on quaternized chitosan (QCS) and tannic acid. It also showed the ability to reduce free radicals. In vivo studies on diabetic rats have shown the suppression of inflammation and acceleration of collagen deposition in skin defects. The multifunctional QCS/TA/Fe hydrogel developed by Guo et al. [174] as a wound dressing for the closure and healing of wounds demonstrated antibacterial properties attributed to its responsiveness to NIR (temperature even increased by 60 °C). The photothermal effect was achieved through the presence of TA/Fe^3+^, and it increased with the increased Fe^3+^ content. In vivo study results indicated that this multifunctional hydrogel dressing effectively closed and healed wounds, eliminating *Staphylococcus aureus* infection, promoting angiogenesis, reducing the inflammation, and decreasing the secretion of various pro-inflammatory cytokines. Su et al. created a PTCPP hydrogel. Its liquid formula turned from sol to gel state at around 30 °C [175]. In vitro antibacterial results showed that the bactericidal rates of PTCPP against *Staphylococcus aureus* and *Escherichia coli* under NIR irradiation were 99.994% and 99.91%, respectively. In in vivo experiments, PTCPP adapted to the shape of the wound, showing good adhesion properties and promoting the healing of infected wounds. A simple one-pot synthesis procedure was utilized to prepare self-adhesive hydrogels composed of poly(acrylamide) (PAM), naturally derived chitosan (CS), and tannic acid/ferric ion chelates (TA@Fe^3+^) [176]. The impressive near-infrared (NIR) photothermal conversion capabilities of TA@Fe^3+^ conferred the excellent antibacterial characteristics to the hydrogels, eliminating the necessity of antibiotics use. This has been confirmed through antibacterial experiments conducted both in laboratory settings (in vitro) and within living organisms (in vivo). Moreover, TA@Fe^3+^ exhibited favorable compatibility with fibroblasts cells, promoting cell attachment, proliferation, and the differentiation of the cells. This acceleration of these processes led to the faster closure of skin wounds and the maturation of tissues. A hydrogel was designed by Laurano et al. [177] that exhibited an ROS-stimulated release of gallic acid. In consequence, the reduction in ROS concentration mediated by gallic acid activity was observed. The hydrogel also had the ability to change consistency (from liquid to gel) in response to temperature changes (specifically to a temperature of 37 °C). Meanwhile, in response to lowering the temperature (up to 3 °C), the hydrogels turn back into a liquid state. A hydrogel composed of two components: gallic-acid-functionalized hyaluronic acid (HAGA) and hyaluronic acid methacrylate (HAMA) showed the ability to swell in an acidic environment while remaining stable in a neutral environment. Thus, the hydrogel exhibited pH-responsiveness [178]. This hydrogel also exhibited a response to the temperature. After 30 min of incubation at 37 degrees, the cut pieces of the hydrogel were able to rejoin. This indicates the self-healing ability of this hydrogel. Moreover, the presence of gallic acid increased the adhesiveness of the hydrogel to tissues, thus enhancing the probability of wound healing acceleration. Yang et al. synthesized a very complex hydrogel based on resveratrol, PEG, CNF (cellulose nanofibrils), and PVA, crosslinked by borax. The resulting hydrogel exhibited a 2.33 times greater release of the resveratrol under acidic pH conditions compared to a neutral pH [92]. Due to the presence of resveratrol, the hydrogel exhibited significant antibacterial and antioxidant properties with beneficial influence on the wound healing process. Ma et al. developed an HPCHC/TA/Fe smart hydrogel with dual stimuli responsiveness (pH and temperature) [179]. The presence of a TA addition in the hydrogel structure acted as a crosslinker to enhance the mechanical properties of the hydrogel and acted as an antibacterial agent. Such hydrogel exhibited a pH-dependent TA release process that was responsible for the antibacterial properties of hydrogel against *Escherichia coli* and *Staphylococcus aureus*. The HPCH/TA/Fe hydrogel precursor solution, prior to gelation at low temperatures, can be injected onto the wound site to fill irregular defects, rapidly forming a gel under physiological conditions. Additionally, in a mouse wound model, it demonstrated the remarkable ability to accelerate wound healing without scars. Increasing graphene oxide (GO) content in hydrogels designed by Alarjan et al. [180] slowed down biodegradation due to complex polymerization. However, it concurrently enhanced mechanical strength and hydrophilicity. The pH-sensitive swelling observed in buffer and non-buffer solutions indicates the hydrogels’ suitability for controlled drug release. Thus, these hydrogels, with a higher GO content, can be employed for controlled curcumin release. Such constructs also possessed antibacterial activity against *Staphylococcus aureus*, *Escherichia coli*, and *Pseudomonas aeruginosa*.

### 4.2. Hydrogels Loaded with Peptides, Polypeptides, and Proteins

With more than one hundred products approved by the US Food and Drug Administration (and many more being actually developed), polypeptide/protein-based therapeutics have gained significant attention in all areas of medicine, including cancer therapies, inflammatory diseases, vaccines, and diagnostics. Polypeptides and proteins can provide highly specific and complex functions that are often unable to be provided by small synthetic compounds, including catalyzing desired biochemical reactions, participating in the formation of membrane receptors and channels, and transporting molecules providing intracellular and extracellular scaffolding support [206]. Therefore, polypeptides and proteins have always been widely studied as therapeutic agents for the treatment of various human diseases [207]. However, their physicochemical properties often render them difficult to be used as bare therapeutic agents [208]. Their incorporation in a three-dimensional hydrogel structure additionally provides a number of possibilities when it comes to therapeutic outcomes; examples of hydrogel systems loaded with polypeptides and proteins are summarized in Table 4.

Lee et al. synthesized a thermosensitive hydrogel made of a triblock copolymer, PEG–PLGA–PEG containing plasmid TGF-β1 (a protein known for its inhibitory action of autoimmune and chronic inflammatory diseases) and used the obtained material to accelerate diabetic wound healing [181]. The bare and TGF-β1-loaded hydrogels were administered to the wound and it was found that while bare hydrogel is slightly beneficial for re-epithealization at an early stage of healing (1–5 days), significantly accelerated re-epithelializaion is observed in the wound treated with a TGF-β1-loaded hydrogel. Moreover, the accelerated re-epithelialization was accompanied by increased cell proliferation, enhanced collagen synthesis, and more organized extracellular matrix deposition. A commercial wound dressing, Humatrix^®^, was also doped with TGF-β1 but the resulting formulation had little effect when compared with the obtained PEG–PLGA–PEG hydrogels, which shows the importance of the proper choice of the matrix.

Li et al. developed a self-healing hydrogel composed of N-carboxyethyl chitosan (N-chitosan) and adipic acid dihydrazide, which was crosslinked in situ by hyaluronic acid–aldehyde and loaded with insulin [182]. This construct exhibits pH-responsive long-term insulin release, offering an appealing mechanism to reduce glucose levels, making it particularly advantageous for diabetic skin wounds.

Temperature-sensitive and thermoreversible hydrogels based on a thermosensitive polymer, poly-(N-isopropylacrylamide) (PNIPAM), were obtained by a combination of a short peptide (I_3_K) with PNIPAM [105]. An antibacterial peptide G(IIKK)_3_I-NH_2_ (a short cationic helical peptide with confirmed antimicrobial properties [209]) was encapsulated in the hydrogel matrix as a model drug. The fabricated composite hydrogel gave a sustained and controlled linear release of G(IIKK)_3_I-NH_2_ over time. Using the peptide nanofibrils as three-dimensional scaffolds, the obtained thermoresponsive hydrogels can mimic the extracellular matrix and could potentially be used for tissue engineering [210]. It should be mentioned that PNIMAM-based hydrogels enable faster drug release at an elevated temperature (e.g., during the inflammatory state of chronic wounds) and slower delivery at lower temperatures, which results from a low critical solution temperature, close to the body temperature [211]. This makes PNIMAM a very promising hydrogel matrix for use in the treatment of inflamed wounds [35].

Several short peptide-based wound healing systems have been reported in recent years [212]. For example, Wu et al. employed an antimicrobial peptide DP7 (a short twelve amino acid cationic peptide with broad-spectrum antibacterial activities [213]) to create a pH-sensitive hydrogel wound dressing, based on preoxidized dextran as the polymeric matrix [183]. The resulting hydrogel not only inhibited the growth of multidrug-resistant bacteria but it also did not cause an increase in bacterial resistance. To enhance its efficacy, the hydrogel was loaded with ceftazidime for synergistic antibacterial effects. The combined action of DP7, ceftazidime, and oxidized polysaccharides exhibited significant efficacy against a variety of multidrug-resistant *P. aeruginosa* strains. Remarkable wound healing was observed in the in vivo experiment, in both wild type C57 and diabetic mouse models [183], which is attributed to the hydrogel erosion accelerating the release rate of the drugs, which is schematically presented in Figure 5.

Rezaei et al. synthesized thermo-responsive chitosan-based hydrogels loaded with various concentrations of piscidin-1 (a fish-derived 22-amino-acid cationic peptide with potent antimicrobial and antiendotoxin activities [214]) to fabricate an antibacterial wound dressing that is able to treat a resistant *Acinetobacter baumannii*. β-glycerolphosphate disodium salt pentahydrate was used to tune the gelation time of the resulting hydrogels. A total of 16 μg·mL^−1^ of piscidin-1 in the hydrogel was found to be the optimal concentration to provide effective antibacterial activity against resistant clinical isolates of *A. baumannii* and no signs of cytotoxicity for human cells were observed [215].

The above-discussed examples of the DP7 and piscidin-1 peptides show increasing interest in antimicrobial peptides (AMPs) as active ingredients of various hydrogel formulations. AMPs constitute an important part of the innate immune defense system in multicellular organisms and can act in two ways: (i) directly, i.e., kill microbial pathogens (most of AMPs), (ii) indirectly: modulate the host defense system [216,217].*Interesting example of Thymosin β4*

One of the most prospectus polypeptides is thymosin β4 (Tβ4). Being the most abundant, it constitutes 70–80% of the β-thymosin polypeptides initially extracted and identified over four decades ago from the calf thymus. It was later shown to be expressed by multiple cells, including immune, brain, liver, testis, myocardium, and blood cells, except for erythrocytes [218]. This is a classical moonlight protein, showing different biological activities in eukaryotic cells, i.e., angiogenic, anti-inflammatory, and anti-microbial properties. Tβ4 upregulates vascular endothelial growth factor VEGF, promotes endothelial cell migration, tube formation, angiogenesis, and wound healing in vivo [219,220]. Via the downregulation of chemo- and cytokines, it can decrease inflammation [221,222], while in platelets it shows antimicrobial properties [223]. As Tβ4 is highly expressed in platelets and wound fluids, this explains its contribution to wound healing and tissue regeneration [224]. In preclinical models and in patients Tβ4 accelerated the rate of dermal healing when applied directly on the injured site or was given intraperitoneally [225]. When Tβ4 was injected intradermally on the second-degree burn wound site it promoted skin regeneration in mice [226], suggesting its potential therapeutic use in the treatment of severe burns. In addition, Tβ4, especially at the SDKP (serine–aspartate–lysine–proline) region, prevents or can reverse uncontrolled wound healing resulting in fibrosis/scarring via the inhibition of macrophage infiltration and secretion of fibrotic factors (TGF-b, IL-10, CTGF) [227].

Interestingly, in human clinical trials on patients with chronic cutaneous pressure ulcers and venous stasis ulcers accompanied with varicose veins and an open ulceration, the topical application of gel containing Tβ4 increased the rate of complete wound healing [225]. As it is a good candidate for future treatments, there are new developments toward new application modes of this factor to improve its effectiveness in wound healing. Transdermal administration of the encapsulated Tβ4 in the ethosomal gels can improve the percutaneous drug absorption and shorten wound recovery [228]. Future prospectives are also attracted to the possible application of Tβ4 in the regeneration of adult tissues such as the heart that was shown in mice where the peptide enhanced myocyte survival and improved cardiac function after coronary artery ligation [229]. For this application the new solutions were developed for Tβ4 delivery. The injectable collagen–chitosan-based hydrogels loaded with Tβ4 were shown to impact heart regeneration by the stimulation of angiogenesis and migration of epicardial heart cells [230]. The controlled delivery and release of Tβ4 into the infarct area was also achieved by the same type of hydrogels with a beneficial effect on the reduction in heart tissue loss and revascularization [231]. The hydrogel solution provides mechanical support to the host tissue, adapting to the geometry of the ventricular space, and offers a less invasive strategy compared to the scaffold patches or other solutions. A similar approach of the long-term Tβ4 delivery was presented with poly(ethylene glycol) (PEG)-based hydrogels [184] or another version [185]. The active factor is released by proteolytic activity tissue-present metalloproteinases (MMPs). This biomatrix provides a three-dimensional environment that is desired when considering the regeneration of vascular structures and networks.

### 4.3. Hydrogels Loaded with Silver Nanoparticles

Given the growing prevalence of multidrug resistant (MDR) bacteria, nanoparticles (NPs) with inherent antibacterial potential, such as silver (Ag) [232], copper (Cu) [233], gold (Au) [234], zinc oxide (ZnO) NPs [235], and more, have emerged as promising alternatives for bacterial infection treatment and preventing biofilm formation [236]. NPs possess distinct physicochemical, optical, and biological properties that are crucial in biomedical applications [33,34]. They also exhibit complex antimicrobial mechanisms, significantly reducing the likelihood of bacterial drug-resistance development. NPs can penetrate the bacterial cell walls, then positive charges of NPs are linked to negatively charged sectors at the surfaces of bacteria. As a result, hydrophobic interactions can lead to holes in bacteria surfaces. In addition, they adversely impact the proton efflux pumps and subsequently, with a modification of the pH range, destroy the membrane’s surface charge [236].

Moreover, NPs with small sizes and large surface areas, including polymeric, liposomal, lipid-based, and inorganic NPs, have the capacity to transport and release therapeutic agents at wound sites. Furthermore, these NPs can be integrated into a variety of wound dressing systems to enhance the safety and effectiveness of infected wound treatment [237]. Consequently, the development of intelligent nanomaterial-based hydrogel carriers capable of responding in a controlled manner to the specific microenvironmental stimuli created by bacterial infections or external stimuli is an effective strategy for the design of wound dressings. These stimuli include an acidic pH, excess of ROSs, bacterial-secreted toxins and enzymes, light, heat, and magnetic fields. So far, these intelligent nanocarriers undergo extensive research and hold the potential for further integration into wound dressings to enhance the wound healing process [87,238].

Among the different metal NPs, silver nanoparticles (AgNPs) are widely employed as bactericidal agents in the treatment of burns and various types of ulcers to prevent bacterial infections [238,239,240,241]. This is due to their unique surface properties, characterized by a high ratio of surface atoms to inner surface atoms and elevated surface energy, along with their small size, resulting in a substantial specific surface area [242]. These features enable AgNPs to disrupt the membrane structure and hinder enzyme activity in bacteria. AgNPs adhere to the bacterial cell membrane, subsequently releasing silver ions or intact nanoparticles into the bacterial cells. Within these cells, they interact with phosphorus and sulfur groups present in proteins and DNA, effectively exerting their antibacterial effects [243]. AgNPs are used in wound healing due to their remarkable anti-inflammatory and antibacterial properties. When AgNPs come into contact with the wound site, they interact with bacterial cells, disrupting their cell membranes and interfering with their metabolic processes. This action helps prevent or reduce infections at the wound site, which is a crucial aspect of efficient wound healing. Moreover, AgNPs exhibit anti-inflammatory effects by reducing immune cell activation and the release of pro-inflammatory cytokines. They also modulate neutrophil activity, helping to control excessive inflammation and creating a more conducive environment for the wound healing process [244,245,246].

The utilization of natural substances obtained from plants and microorganisms for the biosynthesis of nanoparticles has gained significant interest, primarily driven by the growing need to develop environmentally sustainable and non-toxic approaches. This method, which relies on renewable materials and avoids the use of hazardous chemicals and environmentally unfriendly solvents, renders the nanoparticles more suitable for various biomedical applications. Table 5 presents the latest advancements in stimuli-responsive hydrogels for wound healing that are enriched with AgNPs.

Srikhao et al. reported CMS/PVA–H hydrogels based on the idea of AgNP incorporation into cassava starch modified by carboxymethylation (CMS), mixed with PVA and TA. NIR stimulation of the fabricated hydrogel resulted in the generation of elevated temperatures from 21.5 to 31.7 °C, demonstrating enhanced photothermal capabilities [187]. The addition of (TA-reduced) AgNPs to the hydrogel also allowed a pH-responsive release of TA as a therapeutic agent. The increased levels of loaded AgNPs enhanced the antibacterial and mechanical properties of hydrogel. The obtained nanocomposite hydrogel showed remarkable potential for future applications in wound dressings. Haidari et al. developed a facile synthetic procedure to polymerize methacrylic acid (mAA) in combination with acrylamide (AAm) crosslinked with N,N’-Methylenebisacrylamide (MBMa) to establish a sensitive pH-responsive hydrogel delivery system for AgNPs [188]. The prepared hydrogel restricts the release of AgNPs at an acidic pH (pH = 4) but substantially amplifies it at an alkaline pH (pH = 7.4 and pH = 10). This facilitates a controlled AgNP release and exhibits strong antibacterial properties against both Gram-negative and Gram-positive bacteria, all while remaining non-toxic to mammalian cells. A dual-responsive hydrogel (pnipam–PAA–AgNPs) was prepared by the Haidari research group [189]. They achieved this by crosslinking N-isopropyl acrylamide with acrylic acid and incorporating ultrasmall AgNPs. This hydrogel operates by adapting to the physiological conditions of an infected wound, enabling the “on-demand” release of ultrasmall AgNPs based on the wound’s pathological state. It contains a pH-responsive component that swells as the pH increases, promoting the gradual release of AgNPs. It responds to changes in both pH and temperature, by transitioning from a hydrophilic to a hydrophobic state at a temperature near 37 °C. In infected wounds, where the pH is higher, the release of Ag^+^ ions is accelerated, effectively eliminating bacteria and supporting wound healing. Moreover, the Pnipam–PAA–AgNPs hydrogel maintains its ability to respond to body temperature regardless of the pH conditions in the wound environment. This is advantageous in the context of infection treatment and wound healing.

Du et al. [190] designed a composite Ag_2_S quantum dot/mSiO_2_NPs hydrogel (NP hydrogel) with an antibacterial ability. It was constructed by incorporating Ag_2_S quantum dots (QDs) modified by mesoporous silica (mSiO_2_) into the network structure of 3-(trimethoxylmethosilyl) propyl methacrylate based on free radical polymerization. The hydrogel demonstrated remarkable performance, with a photothermal conversion efficiency of 57.3% when exposed to 808 nm near-infrared (NIR) light. Moreover, the hydrogel released silver ions (Ag^+^) in a controlled manner in response to NIR laser-induced volume changes, enhancing its antibacterial properties. It efficiently eliminated 99.7% of *Escherichia coli* and 99.8% of methicillin-resistant *Staphylococcus aureus* (MRSA) within just 4 min under NIR laser exposure. It was also responsive to photodynamic therapy (PDT), generating reactive oxygen species (ROS) upon NIR light exposure, further aiding bacterial eradication. Thermosensitive, injectable nanocomposite hydrogels (Pluronic/HA/CSE/Ag) containing AgNPs demonstrated good mechanical properties with a gelation temperature close to the body temperature, thus allowing for the easy possibility of an application. They exhibited antibacterial activity toward Gram-positive and Gram-negative bacterial strains and allowed for a facilitated accelerated wound closure and regeneration process [168].

Methylcellulose (MC) and AgNPs-loaded hydrogels were prepared and crosslinked with citric acid (CA) at three different crosslinking degrees: low (MC-L), medium (MC-M), and high (MC-H). Pristine hydrogel (MC) was used as a control. All hydrogel variants at 25 °C exhibited a rapid swelling behavior, reaching a high degree of swelling (SR = 4000–6000%). Subsequently, they dissolved within 72 h, irrespective of the pH. This suggests that at a temperature lower than the phase transition temperature (T_t_), these hydrogels were in a liquid state and underwent rapid dissolution in the aqueous environment. Highly crosslinked hydrogels (MC-H) displayed remarkable pH-responsive characteristics, mainly due to selective hydrolysis in an alkaline environment. As an accompanying phenomenon, a significant release of AgNPs was detected (several times higher at pH 12 than at pH 4). Temperature also affected the properties of described hydrogels because they exhibited a significantly higher swelling ratio at lower temperatures (25 °C) than at higher ones (50 °C). MC-H hydrogels were identified as a promising approach for in-situ synthesis of AgNPs, followed by pH-triggered release. This platform seems to be the potential solution to effectively regulate pathogen growth in chronically infected wounds characterized by an alkaline pH environment [15].

A bactericidal nanocomposite was obtained using Ag nanoparticles/phosphotungstic acid–polydopamine nanoparticles (AgNPs/POM–PDA). The final multifunctional wound dressing was obtained by embedding the resulting nanocomposite into the chitosan–gelatin hydrogel [191]. Ag^+^ release from the hydrogel took place under NIR light irradiation. That process was responsible for excellent synergistic anti-bacterial activity against Gram-negative *Escherichia coli* and Gram-positive *Staphylococcus aureus*. The synergistic effect of the simultaneous presence of AgNPs/POM–PDA nanocomposites and CS/GE hydrogel in the final formula remarkably accelerated wound healing in vivo due to the excellent biocompatibility, hydroabsorptivity, and breathability of the hydrogel. The in vivo infectious wound healing test showed that the obtained multicomponent hydrogel-based scaffolds promoted wound healing by inhibiting wound infection and reducing the inflammatory response.

### 4.4. Hydrogels Loaded with Antibiotics and Drugs

Although stimuli-responsive systems can deliver drugs in a controlled manner, the antibiotic-based strategy proves inefficient in the long run, ultimately leading to undesirable effects. This includes increasing local drug concentration at infection sites, antibiotic accumulation in healthy host tissues, the risks of toxicity, and exposure of commensal microflora to sub-lethal antibiotic doses [236]. The biggest danger that systems based on antibiotics face is the risk of the potential development of antimicrobial resistance associated with prolonged exposure to these substances. Moreover, biofilm-forming bacteria are less prone to the action of the human immune system due to the formation of a mechanical barrier as well as antiphagocytic properties of the resulting biofilm, limiting penetration and further action of antimicrobial agents. Secondly, the biofilm environment enables bacterial communication and thus promotes phenotypic changes that are enhanced by the use of antibiotics [247,248]. Besides contributing to the development of resistance, the use of topical antibiotics may also trigger adverse effects, including delayed hypersensitivity reactions, superinfections, and contact dermatitis [249]. In Table 6, the examples of the drug-loaded stimuli-responsive hydrogel formulas are presented.

Zhao et al. created an ROS-scavenging hydrogel designed to eliminate high ROS levels in wound sites [192]. This PVA-based hydrogel was an ROS-responsive TPA linker. Such construct was loaded with mupirocin (MP) (antibiotic) and a tissue-regenerating growth factor (GM-CSF). It showed antibacterial activity and effectively lowered intracellular ROS levels. Additionally, it decreased the secretion of pro-inflammatory factors, controlled macrophage behavior, stimulated the formation of new blood vessels and collagen, and markedly enhanced wound healing capabilities. The study utilized a freeze–thaw method to develop a pH-responsive hydrogel named FTS-G@PC, which is composed of polyvinyl alcohol (PVA) and chitosan (CS), with gentamicin (GS) incorporated and crosslinked within. This hydrogel is exceptionally biocompatible due to its use of natural wood and the physical crosslinking achieved through repeated freezing and thawing. Additionally, gentamicin was released in a weakly acidic pH and enhanced antibacterial capabilities, reducing bacterial growth and increasing mortality rates, especially against *Staphylococcus aureus* and *Escherichia coli* [193]. The uniqueness of this hydrogel lies in its biocompatibility, owing to the use of natural wood and the physical crosslinking achieved through freeze–thaw cycles.

Rezaei et al. [194] proposed the pH-sensitive vancomycin-loaded silk fibroin–sodium alginate nanoparticles (SF–SANPs) embedded in a poly(N-isopropylacrylamide) (PNIPAM) hydrogel containing epidermal growth factor (EGF) for the treatment of chronic burn wound infections. Vancomycin exhibited a pH-dependent release behavior from the nanoparticles with a higher release rate in an alkaline pH compared to the neutral pH values. Hu et al. [195] proposed a dual-responsive hydrogel system (Hydrogel@AM&MIC and Hydrogel@NAP&MIC) by grafting phenylboronic acid to the side chain of the alginate polymer. By grafting phenylboronic acid onto the alginate polymer’s side chain, a highly specific hydrogel responsive to a low pH and high ROS levels was obtained. The hydrogel was endowed with antibacterial and anti-inflammatory properties respectively via the effective assembly of amikacin (antibiotic) and naproxen (anti-inflammatory drug) preloaded into the micelles.

Niyompanich et al. designed poloxamer hydrogels loaded with the gentamicin sulfate, which exhibited antibacterial properties against *Escherichia coli*, *Bacillus cereus*, *Staphylococcus aureus*, and MRSA [196]. Temperature-sensitive NIPAM–CG/GM hydrogels [197] showed strong mechanical properties and an excellent drug loading capacity. Its phase transition closely matches the human body temperature, facilitating efficient drug release. Additionally, the hydrogel effectively prevents microbial invasion in wounds and ensures a moist conducive environment for healing without harmful bacteria.

Sprayable ZnMet-PF127 developed by Liu et al. [198] was used to evenly cover the surface of an irregular skin defect. The application of ZnMet-PF127 promoted granulation tissue formation, collagen deposition, new vessel formation, and inhibited ROS accumulation and inflammation. It also showed antibacterial activity against *Staphylococcus aureus* or *Escherichia coli*.

## 5. Conclusions and Perspectives

This review presents and discusses the-state-of-the-art discoveries regarding stimuli-responsive hydrogels used in wound healing and skin tissue regeneration. Over the last few years, a plethora of various additives (e.g., polyphenols, polypeptides, nanoparticles) have been investigated and promising results have been obtained, as discussed in this review. However, the use of new types of additives to support the wound healing process could lead to significant breakthroughs in this field; a very good example is thymosin β4 discussed in Section 4.2. Despite these significant achievements, there are still important issues that need to be addressed in the future.

Firstly, ensuring high biocompatibility and cytocompatibility not only of the hydrogel itself but also of its degradation products needs to be addressed. For this reason, the development of hydrogels based on natural hydrogelators such as hyaluronic acid, chitosan, gelatin, and collagen is often considered as a first choice.

Secondly, despite great progress being made in the targeted release of bioactive agents/drugs, this field requires further clarification to achieve effective therapies that could be translated to the clinic. Some of the challenges here are: (i) the proper dosage for optimal treatment, (ii) identification of the hydrogel matrix influence on the releasing drug side effects as well as on therapeutic outcomes, and (iii) the precise control over the drug release profile with the minimalization of the unwanted drug-leakage during transportation and avoidance of possible off-target.

Thirdly, new chemical and physical crosslinking approaches better mimicking in vivo dynamic behavior are needed to be developed to broaden hydrogels’ applications; those approaches may include inter alia click chemistry or enzymatic reactions. The projected routes should also focus on the fabrication of pH-responsive hydrogels that are more effective at an alkaline pH, given that the chronic wound environment is alkaline, but most pH-responsive hydrogels currently appear more friendly to acidic environments.

In the future, stimuli-responsive hydrogel-based dressings may provide an excellent control over the wound healing processes, and if integrated with a miniaturized sensor system, they will enable the tailoring of therapeutic strategies. It is worth mentioning that smart hydrogels integrated with sensors are actually conceptualized to deliver real-time information about the wound healing process. Real-time monitoring of the wound healing process is of paramount importance as many related processes and parameters dynamically change, which makes it difficult to develop a dressing that could simultaneously meets the needs of the entire healing process.

Finally, integration with other functional ingredients (e.g., hemostatic, conductive, or adhesive materials) will certainly bring the potential clinical application much closer. This translational approach requires the concerted endeavors of researchers and clinicians in this booming field.

## Figures and Tables

**Figure 1 materials-17-00278-f001:**
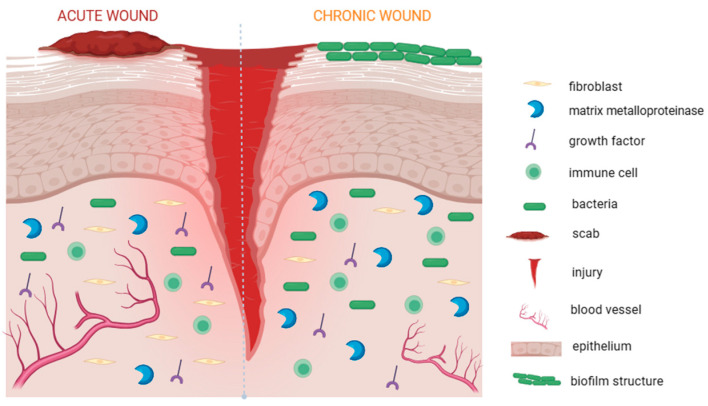
Differences between acute and chronic wounds, created with BioRender.com (accessed on 4 December 2023).

**Figure 2 materials-17-00278-f002:**
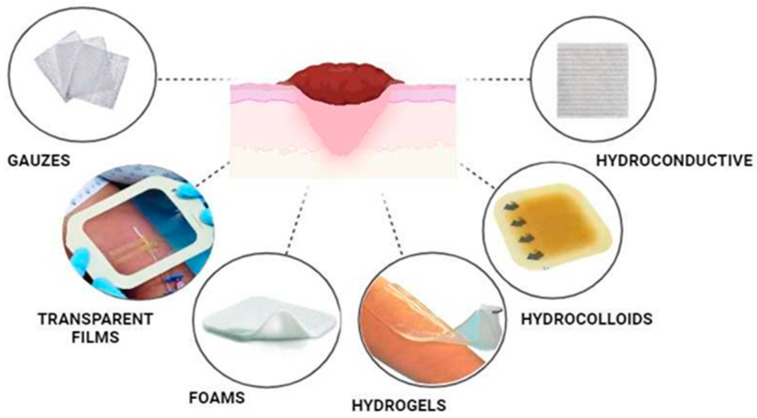
Types of wound dressings, created with BioRender.com (accessed on 4 December 2023).

**Figure 3 materials-17-00278-f003:**
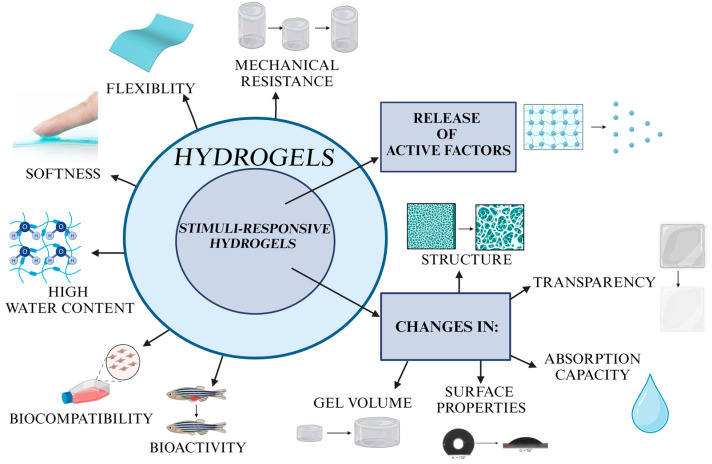
Advantages of stimuli-responsive hydrogels, created with BioRender.com (accessed on 4 December 2023).

**Figure 4 materials-17-00278-f004:**
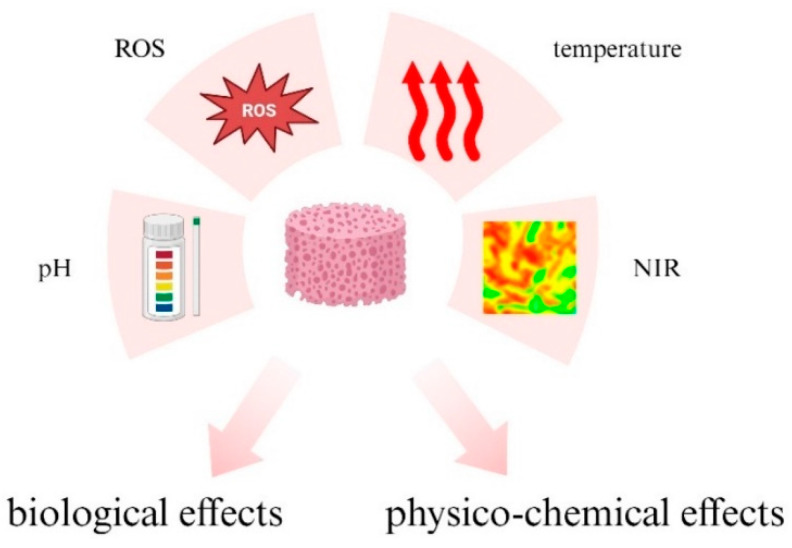
Various stimuli triggering the hydrogels’ response, created with BioRender.com (accessed on 4 December 2023).

**Figure 5 materials-17-00278-f005:**
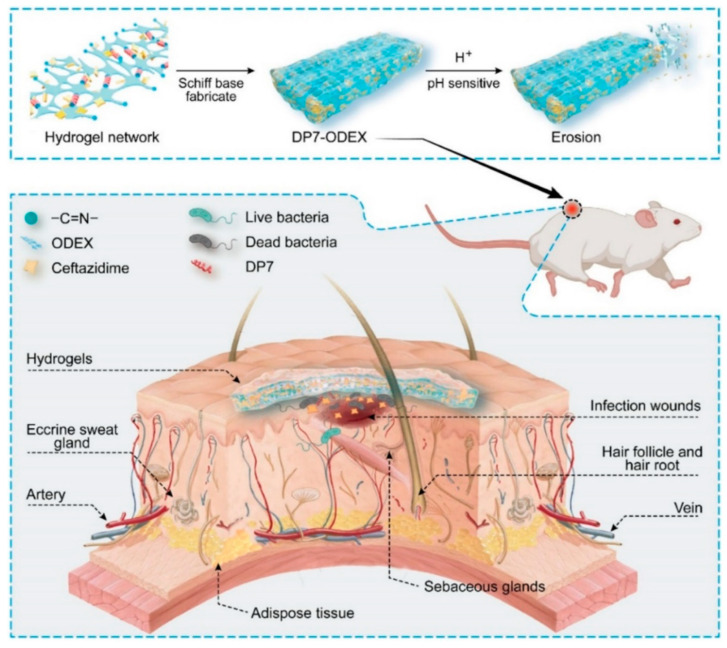
pH-sensitive dextran-based hydrogel erosion accelerating the release rate of the drugs: DP7 peptide and ceftazidime. Reprinted with permission from [183]. Copyright (2022) with permission from Elsevier. *Please note that the correct name is Adipose tissue, not Adispose tissue*.

**Table 1 materials-17-00278-t001:** Examples of stimuli-responsive agents.

Composition	Stimuli Response Agent	Stimuli Mechanism	Material’s Properties	Ref.
Dodecyl, chitosan, WS_2_ nanosheet, ciprofloxacin	WS_2_ nanosheets	WS_2_ nanosheets generated heat upon exposure to near-infrared (NIR) light → triggering the release of the antibiotic at the wound site	Injectable, self-adapting, and rapidly molding hydrogels with good tissue adherence and antibacterial potential	[163]
AuNPs, Pluronic^®^ F127, hydroxypropyl methylcellulose (HPMC)	Pluronic^®^ F127	Stiff gel formation when temperature increased from 4 °C to 32–37 °C	Improved bioavailability, skin permeation, antibacterial and anti-inflammatory activity of the prepared AuNPs’ thermoresponsive gels, burn wound treatment potential	[164]
Gelatin and chondroitin sulfate	Chondroitin sulfate	Tissue adherence at 37 °C, diminished at low temperatures (20 °C), enabling it to detach effortlessly from the tissue	Injectable self-healing bioadhesive, underwater adhesive properties, tissue adhesive and sealant for the closure of bleeding wounds	[165]
Catechol-modified quaternized chitosan, poly(d,l-lactide)-poly(ethylene glycol)-poly(d,lalactide) (PLEL)	PLEL	The temperature-dependent transition of PLEL solution from a reversible sol at 25 °C to a gel at 37 °C	Injectable, thermo-sensitive adhesive hydrogel with promoting wound healing ability, biocompatibility, and bioactivity through in vivo degradation, stimulated endothelial cell migration, and angiogenesis	[166]
Galactose-modified xyloglucan (MxG) and hydroxybutyl chitosan (HBC)	Galactose-modified xyloglucan	Gelation temperature and time can be modulated via adjusting the MxG/HBC ratio	The composite hydrogel could effectively prevent repeated adhesion after adhesiolysis, promote wound healing, and reduce scar formation	[167]
Pluronics, hyaluronic acid, corn silk extract, and nanosilver	Pluronics,	The viscoelastic parameters varied within the temperature range of 25 to 40 °C	Hydrogel with antibacterial activity toward Gram-positive and Gram-negative bacteria	[168]
Collagen (COL), guar gum (GG), poly(N-isopropylacrylamide) (PNIPAM), graphene oxide (GO)	PNIPAM and GO	Phase transition after human body temperature contact; thermosensitive, NIR responsive	Hydrogel with fast self-healing properties, super-ductile, injectable, remoldable, conductive, and skin wound-healing acceleration properties	[169]

**Table 2 materials-17-00278-t002:** Key characteristics of stimuli-responsive hydrogel matrices loaded with active substances.

Hydrogel-Modified Substance	Main Characteristics of Modified Matrices	Refs.
Polyphenols	mechanical strength, structural integrity, adhesion, high elasticity, self-healing properties, hemostatic properties, antibacterial properties, antioxidant properties, anti-inflammatory properties	[70,92,123,173,174,175,176,177,178,179,180]
Peptides, polypeptides, and proteins	Biocompatibility, regeneration processes, stimulation, antibacterial properties.	[181,182,183,184,185,186]
Silver nanoparticles	Antibacterial properties, anti-inflammatory properties, stability, durability.	[15,168,187,188,189,190,191]
Antibiotics	Antibacterial properties	[192,193,194,195,196,197,198]

**Table 3 materials-17-00278-t003:** Polyphenol-loaded stimuli-responsive wound dressing hydrogels.

Hydrogel Composition	Stimuli	Effects	Ref.
Gelatin (Gel), tannic acid (TA) Gel/TA	pH	pH-dependent release of TA.	[70]
Phenylboric acid-modified polyphosphazene (PPBA), tannic acid (TA), poly(vinyl alcohol) PPBA-TA-PVA	ROS	ROS-dependent release of TA (scavenging of 2,2-diphenyl-1-picrylhydrazyl (DPPH) radicals and OH radicals in vitro) ROS-responsive degradation.	[173]
Quaternized chitosan (QCS), tannic acid (TA) QCS/TA	ROS	Self-healing properties, free radical-scavenging activity due to TA presence.	[123]
Physical crosslinked quaternized chitosan (QCS), tannic acid (TA), ferric iron Fe(III) QCS/TA/Fe	NIR	Antibacterial activity induced by NIR-stimulated modified hydrogels.	[174]
Polydopamine (P), tannic acid (T), chitosan (C), poloxamer 407/188 (PP) PTCPP hydrogel	Temp. NIR	Sol–gel transition of liquid hydrogel formulation at around 30 °C, significant enhancement of hydrogel’s antibacterial activity after NIR irradiation.	[175]
Poly(acrylamide) (PAM), naturally derived chitosan (CS), tannic acid/ferric ion chelates (TA@Fe^3+^) PAM/CS/TA@Fe^3+^	NIR	In vivo and in vitro antibacterial activity to prevent microbial infection after NIR stimulation.	[176]
Hyaluronic acid (HA), poly(ether urethane), (D-DHP407), gallic acid (GA), HA/D-DHP407-GA	ROS Temp.	Reduction in intracellular ROS level due to ROS-induced GA release, sol–gel transition of liquid hydrogel precursor in response to temperature changes (37 °C).	[177]
Gallic-acid-functionalized hyaluronic acid (HAGA), hyaluronic acid methacrylate (HAMA) HAGA/HAMA hydrogel	pH Temp.	Swelling under acidic conditions and stability at neutral and basic pH. Self-healing ability at 37 °C and increased hydrogel-to-tissue adhesion due to gallic acid presence.	[178]
Resveratrol (RSV), polyethylene glycol (PEG)- cellulose nanofibrils (CNF) (RPC) Poly(vinyl alcohol) (PVA) RPC+PVA+BORAX→ RPC/PB hydrogel	pH	pH-dependent resveratrol release.	[92]
Hydroxypropyl chitin (HPCH), tannic acid (TA), ferric ion (Fe) HPCHC/TA/Fe	pH Temp.	pH-dependent TA release, temperature-dependent gelation.	[179]
polyvinyl alcohol (PVA), Bacterial cellulose (BC), graphene oxide (GO), curcumin, bacterial cellulose-functionalized-graphene oxide PVA/BC-*f*-GO Crosslinker: tetraethyl orthosilicate (TEOS)	pH	pH-dependent curcumin release.	[180]

**Table 4 materials-17-00278-t004:** Polypeptides-loaded stimuli-responsive wound dressing hydrogels.

Hydrogel Composition	Stimuli	Effects	Ref.
PEG–PLGA–PEG triblock copolymer loaded with TGF-β1 polypeptide	Temp.	Temperature-initiatied re-epithelialization and collagen synthesis	[181]
N-carboxyethyl chitosan, hyaluronic acid–aldehyde, adipic acid dihydrazide, insulin	pH	pH-responsive insulin release	[182]
oxidized dextran, antimicrobial peptide DP7, ceftazidime	pH	pH-sensitive hydrogel erosion accelerating the release rate of the drugs	[183]
PEG-based Tβ4-loaded hydrogels	MMPs	Enzymatic activity-dependent release of Tβ4 mediated by tissue metalloproteinases	[184]
PEG–vinylsulfone-based Tβ4-loded hydrogels	MMPs	Enzymatic activity-dependent release of Tβ4 mediated by tissue metalloproteinases	[185]
Tβ4@TNT–PDA/PVHA	ROS	ROS-dependent Tβ4 release by borate bonding cleavage	[186]

**Table 5 materials-17-00278-t005:** Silver nanoparticles (AgNPs)-loaded stimuli-responsive wound dressing hydrogels.

Hydrogel Composition	Stimuli	Effects	Ref.
Cassava starch modified by carboxymethylation (CMS), poly vinyl alcohol (PVA), CMS/PVA–H tannic acid (TA), Silver nanoparticles (AgNPs) H-AgNPs	NIR pH	NIR-stimulated antibacterial activity pH-responsive TA release	[187]
Methacrylic acid (mAA), acrylamide (AAm), N, N’-Methylenebisacrylamide (MBMa), poly(mAA-co-AAm) hydrogel Mercaptossucinic acid (MSA)-protected AgNPs (MSA–AgNPs) poly(mAA-co-AAm)–AgNPs	pH	pH-dependent AgNP release	[188]
N-isopropylacrylamide (Nipam)+acrylic acid (AAc)→ Pnipam AgNPs Pnipam–PAA–AgNPs	pH Temp.	pH-dependent AgNPs’ release Controlled release and delivery of AgNPs	[189]
N-isopropylacrylamide (NIPAAm), acrylamide (AAm), Ag_2_S quantum dots (Ag_2_SQDs) modified by mSiO_2_ mesoporous silica, (NP hydrogel), 3-(trimethoxylmethosilyl) propyl methacrylate (MPS), Ag_2_S QDs/mSiO_2_ NP–MPS	NIR	NIR laser-induced controlled release of silver ions (Ag+)	[190]
Pluronics F127 and F68, hyaluronic acid (HA), corn silk extract (CSE), AgNPs Pluronic/HA/CSE/Ag	Temp.	Temperature-dependent sol–gel transition	[168]
methylcellulose (MC), citric acid (CA), AgNPs MC/AgNPs nanocomposite hydrogels	Temp. pH	Temperature-induced changes in swelling rate and rheological properties pH-induced changes in swelling rate and rheological properties	[15]
Ag nanoparticles/phosphotungstic acid–polydopamine nano-flowers (AgNPs/POM–PDA), chitosan, gelatin, AgNPs/POM–PDA@ chitosan/gelatin	NIR	Ag^+^ release under NIR light irradiation	[191]

**Table 6 materials-17-00278-t006:** Antibiotic- and drug-loaded stimuli-responsive wound dressing hydrogels.

Hydrogel Composition	Stimuli	Effects	Ref.
N^1^-(4-boronobenzyl)-N^3^-(4-boronophenyl)-N^1^, N^1^, N^3^, N^3^-tetramethylpropane-1, 3-diaminium (TPA), poly(vinyl alcohol) (PVA) TPA + PVA = Hydrogel mupirocin (MP), granulocyte-macrophage colony-stimulating factor (GM-CSF),	ROS	ROS-responsive degradation	[192]
Triplochitin scleroxylon wood (TS), gentamicin (G), polyvinyl alcohol (PVA), chitosan (CS), FTS-G@PC Flexible wood-based hydrogel	pH	pH-responsive gentamicin release	[193]
Poly(N-isopropylacrylamide) (PNIPAM), epidermal growth factor (EGF), silk fibroin–sodium alginate, nanoparticles (SF–SANPs), Vancomycin (VANCO) PNIPAM and EGF/SF–SANPs	pH	pH-responsive vancomycin release	[194]
Hyaluronic acid (HA), boronic acid (BA), HA + BA = hydrogel micelle-loaded amikacin (AM), micelle-loaded naproxen (NAP), Hydrogel@AM&MIC Hydrogel@NAP&MIC	pH ROS	pH-dependent amikacin release ROS-responsive naproxen release	[195]
Poloxamer 188, solution of poloxamer 407, gentamicin	Temp.	Sol–gel transition at around 37 °C	[196]
Vinyl carboxymethyl chitosan (CG), graphene (GM), N-isopropylacrylamide (NIPAM), ciprofloxacin hydrochloride, NIPAM–CG/GM	Temp.	Temperature-dependent drug release	[197]
Pluronic F127 (PF127), complex of zinc and metformin, (ZnMet); ZnMet-PF127	Temp.	Sol–gel transition at around 37 °C	[198]

## Data Availability

The data presented in this study are available on request from the corresponding author.

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
