# Peer review of "Designing Composite Stimuli-Responsive Hydrogels for Wound Healing Applications: The State-of-the-Art and Recent Discoveries"

_materials, 2024, doi:10.3390/ma17020278_

Round 1

Reviewer 1 Report

Comments and Suggestions for Authors

In the title, the first character of each word should be capitalized. Follow the journal guidelines.

In the introduction write about the stimuli-responsive materials.

Lines 23-26, need to rewrite in a better way.

Include fabrication method of stimuli-responsive materials.

Biopolymer extensively used for this purpose please include it.

Lines 131-141, avoid bulletin.

Include the following suggested papers in the manuscript; Carbohydrate Polymers 257 (2022)117633; Gels 2023, 9(6) 451, and Journal of Polymers and the Environment 30 (8) (2022) 3409-3419

Provide a Table and put the information of various types of stimuli-responsive materials for wound healing. In table include also materials performance, fabrication method and other superior characteristics. This will attract to readers.  

Similarly prepare another table for section 4.

Give insight into the global research trends.

Table 1 has limited information. Please expand it.

Arrange section 5 into 2 paragraphs one is conclusions and another is future perspectives. Mark the section 5 title as future perspectives.

Comments on the Quality of English Language

Moderate editing of English language required

Author Response

Reviewer 1.

In the title, the first character of each word should be capitalized. Follow the journal guidelines.

As suggested by the Reviewer, the title has been changed.

In the introduction write about the stimuli-responsive materials.

We devoted the entire chapter 4 to characterize the stimuli-responsive materials, therefore this issue does not appear directly in the introduction; however we added some information at the end of the Introduction section.

Lines 23-26, need to rewrite in a better way.

As suggested by the Reviewer, the abstract was reformulated.

Include fabrication method of stimuli-responsive materials.

As suggested by the Reviewer we provided the examples of fabrication method of stimuli-responsive materials, we also added the new Subsection 3.5 with new Table 1.  Also, we would like to point out that our entire chapter 4 deals with plethora fabrication methods of various stimuli responsive hydrogels.

Biopolymer extensively used for this purpose please include it.

As suggested by the Reviewer, we added information in text under Figure 3.

Lines 131-141, avoid bulletin.

As suggested by the Reviewer, the bullet points have been removed.

Include the following suggested papers in the manuscript; Carbohydrate Polymers 257 (2022)117633; Gels 2023, 9(6) 451, and Journal of Polymers and the Environment 30 (8) (2022) 3409-3419

As suggested by the Reviewer, the suggested papers have been included in the manuscript.

Provide a Table and put the information of various types of stimuli-responsive materials for wound healing. In table include also materials performance, fabrication method and other superior characteristics. This will attract to readers. 

As suggested by the Reviewer we provided the examples of fabrication method of stimuli-responsive materials, we also added the new Subsection 3.5 with new Table 1.  Also, we would like to point out that our entire Section 4 deals with plethora fabrication methods of various stimuli responsive hydrogels.

Similarly prepare another table for section 4.

As suggested by the Reviewer, we added the new Table 2.

Give insight into the global research trends.

As suggested by the Reviewer, we provided additional information about global research trends in Introduction. Moreover, we would like to note that our entire review concerns very current global research trends.

Table 1 has limited information. Please expand it.

As suggested by the Reviewer, Table 1 (after renumeration it is now Table 3) has been expanded.

Arrange section 5 into 2 paragraphs one is conclusions and another is future perspectives. Mark the section 5 title as future perspectives.

As the Reviewer 3 assessed the section 5 as very well written without the need to make any changes, for now, we have left it unchanged because fulfilling the request of Reviewer 1 would be contrary to the assessment of Reviewer 3 and vice versa. We would be grateful if the Reviewer 1 would be able to reconsider her/his request.

Reviewer 2 Report

Comments and Suggestions for Authors

Michalicha et al. discuss the formulations and properties of composite stimuli-responsive hydrogels for wound dressing applications. This review manuscript is well-written and organized. The infographics are well illustrated. Therefore, the manuscript can be accepted for publication in materials after the following minor corrections:

·         In section 3., there is no need to keep writing the full names of reactive oxygen species, lower critical solution temperature, and near infrared after you’ve abbreviated them. The same applied to Tannic acid (lines 441, 445, 449, and 683) and silver nanoparticles (AgNPs) in 661, 676, 699, etc.

·         The names of PVA and PNIPAAM in line 349 should be written in full. Moreover, the full name of PEG should be written in line 348 instead of line 351.

·         In line 351, PEG is not a natural polymer, it’s a synthetic polymer. This must be corrected by removing PEG from the list.

·         Names of bacterial strains must always be written in italics (rectify ‘’S. aureus’’ in line 474) and do so throughout the manuscript.

·         Your referencing style should be consistent; for example, the year must be in bold (check references 1, 3, 9, 15-20, etc.).

·         There are some grammatical typos that must be corrected throughout the manuscript.

Author Response

Reviewer 2.

Michalicha et al. discuss the formulations and properties of composite stimuli-responsive hydrogels for wound dressing applications. This review manuscript is well-written and organized. The infographics are well illustrated. Therefore, the manuscript can be accepted for publication in materials after the following minor corrections:

In section 3., there is no need to keep writing the full names of reactive oxygen species, lower critical solution temperature, and near infrared after you’ve abbreviated them. The same applied to Tannic acid (lines 441, 445, 449, and 683) and silver nanoparticles (AgNPs) in 661, 676, 699, etc.

As suggested by the Reviewer, we corrected all the issues related to abbreviations.

The names of PVA and PNIPAAM in line 349 should be written in full. Moreover, the full name of PEG should be written in line 348 instead of line 351.

As suggested by the Reviewer, we corrected all the issues related to abbreviations.

In line 351, PEG is not a natural polymer, it’s a synthetic polymer. This must be corrected by removing PEG from the list.

The Reviewer is absolutely right; we removed PEG from the list as suggested.

Names of bacterial strains must always be written in italics (rectify ‘’S. aureus’’ in line 474) and do so throughout the manuscript.

The reviewer of course is right; names of bacterial strains are now written in italics in the revised manuscript

Your referencing style should be consistent; for example, the year must be in bold (check references 1, 3, 9, 15-20, etc.).

We revised the citation style for all cited publications and we have fixed the issues raised by the Reviewer.

There are some grammatical typos that must be corrected throughout the manuscript.

As suggested by the Reviewer, we thoroughly have revised the entire manuscript correcting typos we had found.

Reviewer 3 Report

Comments and Suggestions for Authors

The manuscript titled “Designing composite stimuli-responsive hydrogels for wound healing applications: the state-of-the-art and recent discoveries” by Michalicha, A.; et al. is a Review work where the authors deeply discussed about the applicability of hydrogels in the field of wound healing describing the most relevant strategies to make them tunable to external stimuli enhancing their properties. This could be a preliminary step to develop customized targeted-protocols not only suitable in injury treatments, but also expandable for other human malignancies.

However, it exists some points that need to be addressed (please, see them below detailed point-by-point). The most relevant outcomes remarked by the authors can contribute in the growth of many fields like the design of the next-generation of suitable materials for wound healthcare purposes. For this reason, I will recommend the present scientific manuscript for further publication in Materials once all the below described suggestions will be properly fixed.

Here, there exists some points that must be covered in order to improve the scientific quality of the manuscript paper:

1) KEYWORDS. (OPTIONAL) The authors should consider to add the term “drug delivery strategies” in the keyword list.

2) INTRODUCTION. “Wounds-related complication affect over six million people annually in the United States, incurring a cost of $25 billion (USD)” (lines 47-49). The authors should also furnish quantitative information about the worldwide incidence and cost of human wounds [1].

[1] Sen, C.K. Human Wounds and Its Burden: An Updated Compendium of Estimates. Adv. Wound Care 2019, 8, 39-48. https://doi.org/10.1089/wound.2019.0946.

3) “Acute wounds are wounds that (…)” (lines 66-68). Please, the authors should rephrase this sentence in order to avoid redundancies. This comment should be taken into account for the rest of the main manuscript body text.                                              

4) Figure 1 (line 103). Where is this Figure referred in the main manuscript body text?

5) “Numerous (…) gauzes, transparent films, foam dressings, hydrogels, hydrocolloids and hydroconductive dressings” (lines 118-119). Here, the authors should add more information about the formulation of the aforementioned materials particularly in hydrogels [2] and hydrocolloids [3].

[2] Shen, Z.; et al. Advances in Functional Hydrogel Wound Dressings: A Review. Polymers 2023, 15, 2000. https://doi.org/10.3390/polym15092000.

[3] Pele, K.G.; et al. Hydrocolloids of Egg White and Gelatin as a Platform for Hydrogel-Based Tissue Engineering. Gels 2023, 9, 505. https://doi.org/10.3390/gels9060505.

6) CHALLENGES RELATED TO HYDROGEL WOUND DRESSINGS. This section clearly outlines the potential limitations in the use of hydrogels for wound healing materials. “Due to combination of high water content, softness, flexibility, biocompatibility and bioactivity (…) field” (lines 147-148). A small statement should be added about the durability factor of hydrogels and how it can affect the final selection.           

7) VARIOUS STIMULI TRIGERRING THE HYDROGEL’S RESPONSE. “Stimuli identified (…) pH, reactive oxygen species (ROS), temperature and near-infrared radiation (NIR)” (lines 223-225). Please, the authors should not neglect the contribution of ionic strength on the performance of the examined hydrogels [4] (in addition to the above described factors).

[4] Vigata, M.; et al. Deciphering the Molecular Mechanism of Water Interactions with Gelatin Methacryloyl Hydrogels: Role of Ionic Strength, pH, Drug Loading and Hydrogel Network Characteristics. Biomedicines 2021, 9, 574. https://doi.org/10.3390/biomedicines9050574.

8) LOADING STIMULI-RESPONSIVE HYDROGELS WITH ACTIVE SUBSTANCES. In this section, the authors list the existing stimuli that can lead changes according to the hydrogel properties. This section is well-structured but it lacks information about the techniques employed to monitor these variations. “This contributes to improved mechanical strength (…) exceptional properties such as adhesion, high elasticity (…) hydrogels” (lines 424-427). For example, here it may be desirable to indicate tools to assess the mechanical [5] and adhesive [6] properties of the hydrogels when they are exposed to external stimuli factors.

[5] Magazzù, A.; et al. Investigation of Soft Matter Nanomechanics by Atomic Force Microscopy and Optical Tweezers: A Comprehensive Review. Nanomaterials 2023, 13, 963. https://doi.org/10.3390/nano13060963.

[6] Pepelyshev, A.; et al. Adhesion of Soft Materials to Rough Surfaces: Experimental Studies, Statistical Analysis and Modelling. Coatings 2018, 8, 350. https://doi.org/10.3390/coatings8100350.

9) Then, the authors should briefly discuss about the potential reusability of the examined hydrogels once they have sensed the external stimuli and performed their biological action.

10) CONCLUSIONS AND PERSPECTIVES. This section straightforward depicts the most relevant findings gathered in this Review work and also the promising future avenues to pursue this exciting research field. No actions are requested from the authors.

11) REFERENCES. The references are in the proper format style of Materials. No actions are requested from the authors.

Comments on the Quality of English Language

The manuscript is well-written. The authors should recheck it in order to polish final details susceptible to be improved. 

Author Response

Reviewer 3.

The manuscript titled “Designing composite stimuli-responsive hydrogels for wound healing applications: the state-of-the-art and recent discoveries” by Michalicha, A.; et al. is a Review work where the authors deeply discussed about the applicability of hydrogels in the field of wound healing describing the most relevant strategies to make them tunable to external stimuli enhancing their properties. This could be a preliminary step to develop customized targeted-protocols not only suitable in injury treatments, but also expandable for other human malignancies. However, it exists some points that need to be addressed (please, see them below detailed point-by-point). The most relevant outcomes remarked by the authors can contribute in the growth of many fields like the design of the next-generation of suitable materials for wound healthcare purposes. For this reason, I will recommend the present scientific manuscript for further publication in Materials once all the below described suggestions will be properly fixed. Here, there exists some points that must be covered in order to improve the scientific quality of the manuscript paper:

1) KEYWORDS. (OPTIONAL) The authors should consider to add the term “drug delivery strategies” in the keyword list.

As suggested by the Reviewer, we added the proposed keyword.

2) INTRODUCTION. “Wounds-related complication affect over six million people annually in the United States, incurring a cost of $25 billion (USD)” (lines 47-49). The authors should also furnish quantitative information about the worldwide incidence and cost of human wounds [1].

[1] Sen, C.K. Human Wounds and Its Burden: An Updated Compendium of Estimates. Adv. Wound Care 2019, 8, 39-48. https://doi.org/10.1089/wound.2019.0946.

As suggested by the Reviewer, we expanded this part of Introduction (see lines 51-68) as well as we cited the suggested excellent paper.

3) “Acute wounds are wounds that (…)” (lines 66-68). Please, the authors should rephrase this sentence in order to avoid redundancies. This comment should be taken into account for the rest of the main manuscript body text.                                              

As suggested by the Reviewer we corrected the indicated phrase as well as we revised the entire manuscript to improve it quality.

4) Figure 1 (line 103). Where is this Figure referred in the main manuscript body text?

The Figure 1 was accordingly referenced in the revised manuscript.

5) “Numerous (…) gauzes, transparent films, foam dressings, hydrogels, hydrocolloids and hydroconductive dressings” (lines 118-119). Here, the authors should add more information about the formulation of the aforementioned materials particularly in hydrogels [2] and hydrocolloids [3].

[2] Shen, Z.; et al. Advances in Functional Hydrogel Wound Dressings: A Review. Polymers 2023, 15, 2000. https://doi.org/10.3390/polym15092000.

[3] Pele, K.G.; et al. Hydrocolloids of Egg White and Gelatin as a Platform for Hydrogel-Based Tissue Engineering. Gels 2023, 9, 505. https://doi.org/10.3390/gels9060505.

As suggested by the Reviewer, we significantly expanded discussion about above mentioned materials (please see the section below the Figure 2). We also cited the suggested references.

6) CHALLENGES RELATED TO HYDROGEL WOUND DRESSINGS. This section clearly outlines the potential limitations in the use of hydrogels for wound healing materials. “Due to combination of high water content, softness, flexibility, biocompatibility and bioactivity (…) field” (lines 147-148). A small statement should be added about the durability factor of hydrogels and how it can affect the final selection. 

In fact we briefly discussed the issues related to durability (lines 226-236 in the revised manuscript). However, based on a reviewer's request, we have expanded this discussion.

7) VARIOUS STIMULI TRIGERRING THE HYDROGEL’S RESPONSE. “Stimuli identified (…) pH, reactive oxygen species (ROS), temperature and near-infrared radiation (NIR)” (lines 223-225). Please, the authors should not neglect the contribution of ionic strength on the performance of the examined hydrogels [4] (in addition to the above described factors).

[4] Vigata, M.; et al. Deciphering the Molecular Mechanism of Water Interactions with Gelatin Methacryloyl Hydrogels: Role of Ionic Strength, pH, Drug Loading and Hydrogel Network Characteristics. Biomedicines 2021, 9, 574. https://doi.org/10.3390/biomedicines9050574.

As suggested by the Reviewer, we added the concise discussion about ionic strength (please see the section just above the Figure 4). We also cited the suggested reference along with two more references.

8) LOADING STIMULI-RESPONSIVE HYDROGELS WITH ACTIVE SUBSTANCES. In this section, the authors list the existing stimuli that can lead changes according to the hydrogel properties. This section is well-structured but it lacks information about the techniques employed to monitor these variations. “This contributes to improved mechanical strength (…) exceptional properties such as adhesion, high elasticity (…) hydrogels” (lines 424-427). For example, here it may be desirable to indicate tools to assess the mechanical [5] and adhesive [6] properties of the hydrogels when they are exposed to external stimuli factors.

[5] Magazzù, A.; et al. Investigation of Soft Matter Nanomechanics by Atomic Force Microscopy and Optical Tweezers: A Comprehensive Review. Nanomaterials 2023, 13, 963. https://doi.org/10.3390/nano13060963.

[6] Pepelyshev, A.; et al. Adhesion of Soft Materials to Rough Surfaces: Experimental Studies, Statistical Analysis and Modelling. Coatings 2018, 8, 350. https://doi.org/10.3390/coatings8100350.

As suggested by the Reviewer, we mentioned this issue (please see the section just above the Table 2). We also cited the suggested references along with two more references.

9) Then, the authors should briefly discuss about the potential reusability of the examined hydrogels once they have sensed the external stimuli and performed their biological action.

This is an interesting question but due to the intended medical use of the discussed hydrogels for the treatment of wounds, the issue of potential regeneration/reusability is out of the question. This results from both legal and scientific reasons. The former simply do not allow the reuse of used dressing material because it is potentially infectious medical waste, while the latter additionally make it unjustified because the used hydrogel does not have the same features as the new one and it is not possible to reload it with another cargo of active component. In the case of non-medical applications of hydrogels (e.g. environmental remediation), this is undoubtedly an interesting issue, but in this case it is beyond of the scope of this review.

Round 2

Reviewer 1 Report

Comments and Suggestions for Authors

The authors improved the manuscript.

Comments on the Quality of English Language

Nimnoe checking is needed.

Reviewer 3 Report

Comments and Suggestions for Authors

The manuscript was greatly improved after this round of revision. Based on the significance of the shown data I may warmly endorse this work for further publication in Materials journal.